# Applications and Life Cycle Assessment of Shape Memory Polyethylene Terephthalate in Concrete for Crack Closure

**DOI:** 10.3390/polym14050933

**Published:** 2022-02-25

**Authors:** Riccardo Maddalena, John Sweeney, Jack Winkles, Cristina Tuinea-Bobe, Brunella Balzano, Glen Thompson, Noemi Arena, Tony Jefferson

**Affiliations:** 1School of Engineering, Cardiff University, Cardiff CF3 4AE, UK; maddalenar@cardiff.ac.uk (R.M.); winklesj@cardiff.ac.uk (J.W.); balzanob@cardiff.ac.uk (B.B.); jeffersonad@cardiff.ac.uk (T.J.); 2Department of Mechanical and Energy Systems Engineering, Faculty of Engineering and Informatics, University of Bradford, Bradford BD7 1DP, UK; ctuineab@bradford.ac.uk (C.T.-B.); g.p.thompson@bradford.ac.uk (G.T.); 3ARUP Group Ltd., 8–13 Fitzroy St., Bloomsbury, London W1T 4BQ, UK; noemi.arena@arup.com

**Keywords:** PET, shape memory polymer, crack closure, self-healing, civil engineering, crack width, LCA

## Abstract

Shape memory polymer (SMP) products have been developed for application as crack closure devices in concrete. They have been made from PET in the form of both fibres and hollow tubes. Here, manufacturing methods using die-drawing and mandrel-drawing to induce shape memory are reported. The fibre-based devices are incorporated into concrete and, upon triggering, exert shrinkage restraint forces that close cracks in the concrete. The evolution of shrinkage restraint force in the fibres as manufactured was measured as a function of temperature, showing stresses in excess of 35 MPa. Tendons consisting of fibre bundles are incorporated into concreate beams subjected to controlled cracking. When activated, the tendons reduce the crack widths by 80%. The same fibres are used to produce another class of device known as knotted fibres, which have knotted ends that act as anchor points when they incorporated directly into concrete. Upon activation within the cracked concrete, these devices are shown to completely close cracks. The tubes are used to enclose and restrain prestressed Kevlar fibres. When the tubes are triggered, they shrink and release the prestress force in the Kevlar, which is transferred to the surrounding concrete in the form of a compressive force, thereby closing cracks. The Kevlar fibres also provide substantial reinforcement after activation. The devices are shown to be able to partially and fully close cracks that have been opened to 0.3 mm and achieve post-activation flexural strengths comparable to those of conventional reinforced and prestressed structural elements. Finally, a preliminary life cycle assessment study was used to assess the carbon footprint a nominal unit of concrete made with SMPs fibres compared to conventional concrete.

## 1. Introduction

Concrete is one of the most used materials in the built environment, with an annual production of 4.1 Gt in 2019 [1]. Its production accounts for approximately 40% of the carbon emissions attributed to the building sector [2]. The composition and properties of normal strength concrete (C20–C50) made with Portland cement are well-established and its propensity to crack due to its low tensile strength is well understood [3].

Being a relatively brittle material and being very weak in tension, cracks are to be expected when significant tensile stresses are induced. A concrete structural element may be subjected to a combination of stresses which could cause the formation of cracks where and when the tensile strength of the material is exceeded. Cracking in concrete can also be caused by early age thermal effects, structural loading, differential creep and shrinkage, chemical changes within a structural element, and environmental conditions such as earthquakes or floods. Whatever the cause, surface and internal cracks are inevitable during the service life of a concrete structure [4].

Cracks undermine structural integrity and can lead to reduced durability by allowing the ingress of water and other substances [5]. For example, road bridges are vulnerable since de-icing salts can penetrate cracks and accelerate corrosion of the steel reinforcement or prestressing tendons. The associated expansion of the corrosion products leads to further cracking and, in the most extreme cases, to partial or complete collapse [6].

Although an increased effort has been made to design more durable structures with longer service lives, it remains the case that many modern concrete structures require significant maintenance within the first ten to fifteen years of their service lives [7]. Such repairs place a considerable financial and environmental burden on the owners on the structures as well as on society in general [8]. In Europe, 50% of the annual construction budget (2013) was estimated to be spent on rehabilitation and repair of the existing structures [7], and this proportion has increased to 65% in recent years, with a average maintenance expenditure for highway infrastructure of EUR 21 billion in 2014 alone [9].

The above considerations form a strong case for the introduction of technologies that combat cracking in concrete, either by preventing the formation of cracks or stopping their growth.

One way of minimising or controlling the width of cracks in concrete structural elements is by the addition of shrinkage reducing agents and supplementary cementitious materials, such as MgO, fly ash, crystalline admixtures [10,11,12], often combined with steel or plastic (PVA, PP, PAN, PET) fibres [13,14,15,16].

Polymers have played an important role in civil engineering. For example, they are used in fibres, pipes, geotextile materials, and cladding products [17].

Recycled polymer (PET) particles and fibres have also been used in concrete mixes. These have been used to partially replace conventional aggregate and to enhance the resilience of the material [18,19,20]. For example, Ganesh et al. [20] explored replacing a proportion of the fine aggregate in a geopolymer concrete mix with PET granules. They investigated replacements levels from 5 to 15% and showed that a replacement level of 10% exhibited the best mechanical properties. However, the compressive strength was shown to decrease rapidly as the proportion of PET increased beyond 10%. Islam et al. [21] investigated PET particles as a partial coarse aggregate replacement. The polymer aggregate particles were obtained from waste PET bottles in a process that involved shredding, melting and cooling the material to form PET billets, which were subsequently crushed to form the particles. Coarse aggregate replacements levels from 0 to 50% were considered in mixes with three different water-cement ratios (i.e., 0.42, 0.48, and 0.52). The effects of the PET particles on the plastic and hardened properties of the mix were investigated. The study showed that the workability increased with the percentage of PET coarse aggregate, which was attributed to the smooth surface of the particles providing lower ‘inter particle frictional resistance’. The research also found that increasing the PET concentration reduced the compressive strength of the concrete. The explanation proffered for the latter trend was that the low permeability of the plastic aggregate meant that less water was absorbed by the coarse aggregate phase as a whole, leading to more free water in the mix, a larger void ratio in the hardened concrete and weaker bonds between hardened cement paste and aggregate particles. Salhotra et al. [22] investigated the potential benefits of including extruded fibres (derived from recycled PET bottles( in a concrete mix. In their investigation, a proportion of the fibres were coated with silica fume to enhance the bond properties. The effect on the mechanical properties of including these fibres, of length 25 to 75 mm in the volumetric content range of 3 to 12%, was investigated. It was concluded that the best combination of mechanical properties was obtained with the 25 mm long fibres at a volumetric content of 3%. Specifically, it was shown that this length/content combination produced a concrete with an enhanced tensile strength and fracture resistance, with only a marginal reduction in its compressive strength relative to that of a mix with no fibres.

The incorporation of PET into asphalt pavements has also been considered. In Lugeiyamu et al. it was shown that the replacement of 10% of asphalt binder in stone mastic asphalt mixture enhances the resistance against moisture damage and better performance with regards to low and medium temperatures and fatigue loading [23].

In general, the incorporation of recycled PET into particulate composites makes the material more sustainable, gives economic benefits and often enhances engineering properties [24,25,26].

In this paper, we report a series studies undertaken by the authors to develop crack closure systems for concrete structural elements that use components formed from shape memory polymers (SMPs). Previous publications on this work in civil-engineering journals have concentrated on the structural properties of the concrete elements (see Section 3). By contrast, in this paper the focus is on the polymer science and polymer engineering research undertaken to develop the systems. In addition, this paper presents a new life cycle assessment (LCA) on one of the systems (as discussed below). The main challenges in developing viable SMP-based crack closure systems include achieving the relatively high forces necessary to effect crack closure, the need to manufacture them efficiently and at a structural scale, and the development of trigger mechanisms that can be operated remotely at the optimum time. We describe the method of production of PET SMP fibres that produce high shrinkage restraint forces. These are used in concrete in the form of both fibre bundles (tendons) and as single fibres that anchor at knotted ends (kSMPs). We also describe the production of PET SMP hollow tubes which are used to produce hybrid tendon devices containing prestressed Kevlar fibres. All of these devices can be triggered thermally after installation in concrete beams, and then show significant capability to close cracks and allow the concrete element to recover strength and stiffness. Finally, an LCA has been carried out to assess the environmental impact of the kSMP concrete and the results have been compared to those of a conventional concrete.

## 2. Manufacturing Methods for Shape Memory Polymers

### 2.1. Introduction

The stretching and alignment of macromolecules is the physical basis for the operation of SMPs, and also a means of producing polymer with increased stiffness and strength. The latter factor drove the development of polymer product with aligned molecules—the so-called oriented polymers—that began in the 1970s. Spinning processes have been used for many years to produce oriented polymer fibres of high strength and stiffness [27,28]. For cross-sections too large for spinning, other approaches have been developed. Hydrostatic extrusion, in which high pressure forced a solid polymer extrudate through a converging die, was successfully developed [29] and has found recent applications in various specialised processes [30]. Die-drawing, in which material is pulled through dies or over mandrels by a tensile force, has, however, proved a more tractable process.

Die-drawing was originally developed using conical converging dies [31,32] to produce oriented polypropylene in the form of circular section solid rod. Later, the method was generalised to the production of rectangular sections, again using converging dies [33]. Hollow sections can be die-drawn to produce hollow products. This can be carried out either by using a converging die or a diverging mandrel. In the latter case, circular section tubes are produced by drawing over a conical mandrel to increase the tube diameter, inducing molecular orientation in both axial and hoop directions [34,35]. More recently, the die drawing process has been applied to polymer composites [36,37].

The above solid-phase processes all have potential for the manufacture of SMPs. Die-drawing is more attractive than hydrostatic extrusion as it can be adapted as a continuous process.

### 2.2. Production of SMP Fibres from PET

The selection of the polymer to be used in this application is governed by several factors. To be practical, the polymer must be commercially available on a mass scale so as to be economically feasible. It needs to have sufficient strength and stiffness. If it is amorphous, this implies that it must be below its glass transition temperature under service conditions. It must also function as a shape memory polymer. The mechanism for this effect is the recovery of large strain in its amorphous regions, which can be inhibited in semi-crystalline polymer. These considerations lead us to a low crystallinity polymer with a glass transition above room temperature. When used as an SMP, this form of polymer operates by being deformed above its glass transition temperature and the cooled to prevent recovery; recovery then occurs when the polymer is reheated to its ’trigger’ temperature, which is near or at the glass transition. This introduces another limitation on the choice of polymer, as its trigger temperature must be compatible to its use in setting concrete. Practically this limits the glass transition temperature to lower than the boiling point of water. There are few mass-market amorphous polymers with glass transition between 20 and 100 °C. Some grades of polymethylmethacrylate (PMMA) have glass transition in this range, but preliminary experiments gave poor performance as an SMP. PET has a glass transition temperature of around 90 °C and has the advantage of being recyclable, opening up the potential for the use of recycled polymer in these applications. We therefore made this choice of polymer.

We have adopted the die-drawing process to produce oriented fibres from a commercial PET (Dow Lighter C93) obtained as granules. The process was in two stages.

First, isotropic fibres were made by melt extrusion. The extruder was a Killion S1748, with pressure and screw speed set, respectively, at 30 bar and 15 rpm. Temperatures varied from a maximum of 280 °C to 270 °C at the die head. The material exited through a 4 mm internal dimeter circular die at a haul-off speed of 5 m/min. It was then cooled in a glycerol bath at room temperature. The final product was of diameter 1.8 mm.

Next, the material was die-drawn. The isotropic fibre, having cooled to room temperature, was heated by being pulled through a fan-assisted oven of length 1.5 m. The next stage was the die-drawing, as the fibre enters a converging conical die with cone semi-angle 30° and exit diameter 1 mm, maintained at a controlled temperature (see Figure 1). The process is maintained by pulling the fibre at constant speed by means of a caterpillar-type haul-off device. Maximum molecular orientation is associated with highest possible drawing speed and lowest possible temperature. This condition was approached by setting the caterpillar speed at its maximum 1 m/min (equivalent to a mean strain rate of 17 s^−^^1^ in the die) The oven and die temperatures were then both decreased from 80 °C in 1 °C increments until failure or stress whitening of the fibre were observed. This occurred at 75 °C. The oven was then held at this temperature while the die temperature was incremented until an unvoided fibre was produced. The temperatures established using this procedure were 75 °C in the oven and 80 °C in the die. Once the fibre had left the die, it drew further as it cooled in air between the die and the caterpillar. Its final diameter was 0.9 mm, corresponding to an extension ratio of 4.0. Haul-off force was monitored throughout the process and was found to be 80 N at its start, reducing to a value of 50 N as the process became stable These loads correspond to stresses of 99 and 62 MPa, respectively, and are beyond the yield strength of 55 MPa stipulated in manufacturer’s data sheet.

The performance of these fibres in terms of shrinkage restraint force has been assessed using an apparatus consisting of a small chamber containing a single fibre, held outside the chamber by grips, one of which incorporates a load cell. The chamber was heated by compressed air at a controlled temperature and both the force and temperature were monitored (see Figure 2). Temperature was raised at a constant rate, held at a constant level, and then cooled. Results when heated at 5 °C/min to a hold temperature of 92 °C are shown in Figure 3. The hold temperature is greater than the glass transition temperature measured previously [38] in the range of 70–78 °C, and that of 78 °C according to the manufacturer. The choice of a hold temperature of 90 °C has been shown to be highly effective for this material in terms of obtaining a high shrinkage force [38] and this is consistent with the value used here.

### 2.3. Production of SMP Tubes from PET

For SMP devices to be discussed below, PET tubes were manufactured using the same commercial grade of PET and the same Killion extruder as was used for the fibres above. First, tubes were extruded. Operating conditions were set at a screw speed of 70 rpm, pressure of 200 bar, maximum extruder temperature 280 °C, die head temperature of 260 °C and haul-off speed 500 mm/min. The melt was extruded through a circular section die of diameter 14 mm, with an internal pin of diameter 4 mm to create a central circular hole. The final dimensions were outer diameter 13 mm and inner diameter 6 mm.

The tubes were then die-drawn to obtain oriented product. After heating in an oven, 1 m lengths of extruded tubes were drawn through a pair of dies. As shown in Figure 3, a converging conical die contacts the outside surface of the polymer tube and surrounds a diverging 30° conical mandrel inside the tube. The converging die imposes a degree of axial orientation in the tube. The diverging mandrel induces tangential or hoop orientation of the tube. The air temperatures of both the oven and the die were set at 90 °C. The end of the tube emerging from the die was gripped using a haul-off device, operating initially at 40 mm/min, increasing to achieve steady conditions at 300 mm/min. The tube’s average final outer diameter was 8.7 mm, corresponding to an extension ratio of 4.0. This process induces molecular orientation that not only contributes to shape memory, but also increases the axial compressive strength of the tube. This is essential to the operation of the SMP device that will be described below in Section 3.3.

## 3. Applications in Self-Healing Concrete

### 3.1. Introduction

The SMPs described above have been used to create devices that have been used in concrete for civil engineering applications. This has been the result of collaboration between civil engineers and polymer engineers. The performance and effectiveness of the devices have been reported in mainly civil engineering journals where only summary details of the polymer technology were given. The purpose of the present paper is to discuss and report the polymer aspects of the work. However, we consider it necessary to give some summary details of the devices’ effectiveness. This represents the purpose of this section, within which the civil engineering publications are referenced.

### 3.2. SMP Tendons for Crack Healing

A crack closure system based on SMP tendons was proposed by Jefferson et al. (2010), and further developed by Dunn et al. (2011) [39,40]. In the early work, the tendons comprised multiple strips of PET Shrinktite tape. These were tested in small-scale notched mortar beams (120 *×* 25 *×* 25 mm) with an axial void to accommodate the tendon. Each specimen was cast and cured, and then the PET tendon was installed in the void and clamped at the beam’s extremities. Each beam was loaded in flexure to create a crack of known crack mouth opening displacement (CMOD) and then demounted and placed in an oven to release the shape memory potential. It was shown that the activated tendons were able to close serviceability sized cracks in these small plain mortar beams, and that the system provided favourable conditions for autogenic healing [41]. This early work proved the concept of the SMP crack-closure system but was very limited in scale, and relied on oven activation.

In order to scale-up the system and prove its viability for structural elements, larger scale tendons were needed. These tendons were formed from PET fibres, manufactured as described in Section 2.2, with each tendon comprising bundles of 200 fibres [42], as illustrated in Figure 4. Once the potential of the PET fibres to produce the required restrained shrinkage force had been demonstrated, a tendon assembly was developed comprising the fibres, a heat distribution layer, an electrical heating system and a casing (or sleeve) formed from PLA (see Figure 4). The tendon assembly could now be incorporated into a cementitious element during casting and activated remotely when required [42].

Two beam configurations were used, namely one with PET tendons alone and the other with accompanying steel reinforcement. During testing, the surface crack opening displacements were measured using optical microscopy, and a DIC system was used to measure strains on one side of the beam. In addition, a number of plain concrete ‘control’ beams, without tendons or reinforcement, were cast and tested. A typical load-CMOD response of a beam with a tendon (denoted PET) is compared with that of a control specimen in Figure 6. This illustrates the sequence of responses exhibited by the PET specimen during initial loading (1–2), unloading (2–3), activation (3–4), reloading (4–5) and unloading (5–6). It may be seen that CMOD reduced substantially when the tendon was activated, and that this was accompanied by a significant regain of the flexural stiffness. and strength.

PET tendons were used both as the sole means of reinforcement and in conjunction with steel; configurations are shown in Figure 5. Crack opening was measured during testing using optical microscopy and a DIC system using a speckle pattern. The results were analyzed by comparing tendon reinforced beams with controls. One such comparison is shown in Figure 6, where loads are plotted against crack opening. Activation of the tendon results in crack closure, typically from 0.035–0.063 mm before activation to 0.007 mm after activation. The maxiumum applied load in the reloading stage is close of the initial loading stage.

Whilst the multi-fibre tendon assemblies proved able to close cracks in plain concrete specimens and provide some post-activation flexural capacity, the strength was of similar magnitude to the flexural capacity of the plain concrete beam but with substantially lower stiffness. The tests with the combined PET and steel reinforcement showed that the delayed prestress was insufficient to bring about full crack closure. Therefore, research was undertaken to develop a hybrid tendon system that could not only close cracks but also provide substantial levels of post-activation reinforcement and prestressing. The solution developed was the hybrid tendon described in the following subsection.

### 3.3. Hybrid Tendons Using SMP Tube

A hybrid tendon comprises a pretensioned Kevlar core and a restraining SMP tubular sleeve. The die (or mandrel) drawing technique used to form the latter was described in Section 2.3 and produces tubes with both axial and hoop orientation. The molecular orientation produces shape memory effects and also enhanced strength and stiffness while below the trigger temperature. It was found that the balance between axial and hoop orientation produced by the die/mandrel drawing process gave a higher compressive strength than that produced by axial orientation alone. In particular, the compression strength, originally around 50 MPa, was found to be 90–100 MPa, which is enough to prevent the tube from crushing under the load of the pretensioned Kevlar core.

The process used to manufacture the hybrid tendons (shown in Figure 7) is described in references [43,44]. This involves pre-tensioning the Kevlar strands against the restraining sleeve and then tightening the anchorage assemblies. It should be noted that, with this hybrid tendon design, buckling of the sleeve avoided, since the Kevlar inner cable in tension produces a stabilising inner element. The SMP can be successfully activated by direct heating as in [43,45] or by electric current [42,46].

The ability of the system to close cracks is illustrated in Figure 8, which shows results from a series of tests on flexural beam specimens reported in [43]. Figure 9 shows the force-CMOD response from a test on a beam specimen (255 × 75 × 75 mm) with four embedded tendons. This shows that the system not only produces crack closure but gives a substantial post-activation flexural capacity and stiffness, comparable with that of a conventional prestressed beam. Furthermore, it may be noted that there is no significant post-cracking softening in the response. These results shows that hybrid tendons are viable for full-scale structural application.

### 3.4. SMP Knotted Fibre Composite

Additional research considered PET knotted shape memory polymers (PET k-SMP) in the form of filaments manufactured as described above in Section 2.2 [45]. This research benefits of adding mechanical anchorages to the ends of the fibres, providing additional reactional force when the k-SMP fibres were heated in order to contract and close the crack. The mechanical anchorage was tested by means of a modified pull-out test. Concrete blocks were prepared with both knotted or non-knotted fibres and set up such that two concrete blocks were joined by a singular filament, with one block consequently experiencing a force away from the other block. This caused the tendons without a knot to fail at levels between 10 N to 30 N—levels significantly lower than that of steel fibres but which are comparable to polypropylene (PP) and PET fibres. In contrast to these low resistive forces, the k-SMPs were able to withstand forces of 230 N to 290 N—loads which are comparable to the performance of steel fibres.

When casting the specimens (using prismatic moulds) for crack closure, it was decided that two distributions of sample would be made—those with an aligned distribution (AD) where 30 fibres were specifically placed in the bottom half of the specimen where most cracking was expected to occur, and the other sample contained fibres in a random distribution (RD) across the specimen, adding the fibres during the mixing process. The samples then underwent a three-point bending test, creating cracks of 0.3 mm in width using the CMOD clip gauge. It is interesting to note an elasticity within the k-SMP specimens upon unloading where the cracks closed to between 0.08 mm and 0.15 mm immediately upon release. This initial crack closure can be attributed to the elastic modulus of the PET fibres and was seen to stay in effect for the period after, closing the gaps further. After unloading, the samples were placed horizontally to rest for 24 h before subsequently being heated for 3 h, thus activating the k-SMPs. They were then left to air-cool for 12 h, providing opportunity for the cooling k-SMPs to retract further. Crack widths were subsequently measured. As was predicted by the researchers, the AD samples saw smaller cracks but larger proportionate differences in closure. Whereas the RD samples only averaged a crack closure of 45% across the front and back of two specimens, the AD samples averaged 77% crack closure. In addition, some of the AD data saw the k-SMPs completely close the gap as shown in Figure 10.

It is clear that the orientation of the fibres has a strong impact on the performance of the tendons for the closure of cracks and therefore can act as a highly effective method for crack closure. Overall, it was concluded that the system was highly effective and creates opportunity for further research into the combination of these tendons with other self-healing technologies, such as micro-capsules and crystalline admixtures, in order to achieve total crack and stiffness remediation.

## 4. Life Cycle Assessment of k-SMP Fibres Concrete

The goal of the study was to compare, by means of a life cycle assessment (LCA), the potential environmental impacts of the kSMP concrete with those of conventional concrete. The analysis aims in particular to identify scenarios in which the kSMP concrete show advantages over conventional concrete. The assessments include the manufacturing of the two materials. However, use and end-of-life phases are currently out of scope due to lack of details at this stage.

The following assumptions have been taken into account:1 cubic metre of concrete (2400 kg) delivered to Cardiff.Concrete is produced in Rugby (CEMEX, distance to Cardiff: 250 km), using Portland cement Cem II 32.5 MPa, with a cement:sand:coarse aggregate ratio of 1:2:4 and water:cement ratio (*w*/*c*) of 0.5.Knotted SMP fibres are produced at University of Bradford (distance to Cardiff: 420 km) from commercial grade PET granulates (Dow Lighter C93), by melt extrusion followed by die drawing, both at 280 °C [42].In kSMP concrete, fibres are mixed with the concrete at a volume content of 0.82%, partially replacing coarse aggregate, as described in [45].Energy consumption associated with the concrete mixing and the activation system it is assumed negligible for this study.

The LCA was carried out according to ISO standards [47,48]. The functional unit is the manufacturing of 1 m^3^ of concrete. Since the purpose of the study is to compare two different materials, an attributional approach was adopted [49,50,51]. If the materials do prove to be successful from a market and performance perspective, it will be appropriate to follow up with a whole life cycle consequential LCA but, at this stage, such an analysis would be too speculative to be meaningful [52]. The life cycle environmental impacts were assessed using the CML-2001 methodology developed at the University of Leiden [53]. The following midpoint potential impacts were considered: abiotic depletion, acidification, eutrophication, terrestrial ecotoxicity, human toxicity, global warming, ozone layer depletion and photochemical ozone creation.

In accordance with the ISO standard 14044 (2006), normalization has been used to identify the impact categories most significant for the system under analysis. The LCA calculations have been performed under use of openLCA 1.0.3 [54], and the Ecoinvent 3.8 database was used to model the system.

The principal components of the two concretes assessed are listed in Table 1, along with the other key inputs to the inventory analysis.

The environmental impacts for the kSMP concrete are compared with those of conventional concrete in Figure 11 in terms of contribution to the dominant impact categories. For both concretes, the environmental impacts are dominated by human toxicity (HT), global warming potential (GWP), and acidification potential. The kSMP results in higher values for all of the mentioned impact categories due to the manufacturing of the PET.

Results for these three categories for the kSMP and conventional concrete are shown in Table 2 as well as in Figure 12 and Figure 13 for the GWP and HT, respectively, also reporting the individual processes. The dominant role of the cement production is evident from previous graphs with the production of the PET which makes the kSMP concrete a solution with higher environmental impacts compared to the conventional one. This is primarily due to the energy consumption associated with the PET production. It is important to note that this LCA is cradle to gate and does not include the whole life cycle of these products, especially their crack closure activation during the service life. Further investigation needs to be conducted to predict the durability of kSMP concrete and account for in a LCA cradle to grave approach, as well as re-define the functional unit of the LCA assessment [52,55,56]. This means that the real benefit of the kSMP of reduction for maintenance needed is not reflected in this study. However, it will be reported in following assessments.

The crack closure technologies presented contribute to the decarbonization of the concrete industry, by reducing the maintenance and repair and extending the service life of concrete structures. SMP products can be used in applications such as pre-cast concrete factories, where early age shrinkage cracking may compromise the quality and structural integrity of the elements. Another application is within the bridge construction sector, where early detection of damages and their repair is vital to ensure safety and long-term durability. Further, kSMP fibres could be used in engineering structures where regular inspections are challenging, such as oil-well cements and nuclear waste storage encapsulation. In both cases, the heat released by the pump fluid and the nuclear by-products, respectively, can be used to trigger the fibres shape memory effect.

## 5. Conclusions

SMPs are being used to improve concrete structural integrity in a number of ways. High performance PET fibres have been produced by die-drawing and developed to provide high levels of shrinkage restraint force. SMP tubes, with complex molecular orientation fields, have been developed by mandrel drawing to withstand high compressive loads before activation and enable effective prestressed reinforcement devices.

From the research reported in this paper, it may be concluded that the incorporation of PET-based SMP devices into cementitious structural elements provides an effective means of closing cracks that form after casting and curing.

PET tendons alone are able to close cracks in plain mortar and concrete beams but have insufficient strength and stiffness to provide viable levels of prestressing for full-scale structural elements. The level of delayed prestressing afforded by these tendons is also insufficient to have a significant effect on reinforced concrete elements.

Hybrid tendons, comprising bi-oriented PET tubes and a prestressed Kevlar core, are able to close serviceability-sized cracks in structural elements and provide substantial post-activation prestressing and reinforcement.

Knotted PET SMP fibres used in cementitious composites provided a viable solution to effectively close cracks upon thermal activation in both aligned and random distribution, showing crack width reduction up to 77%.

Future research on the SMP-based systems will involve the development of automated self-activation systems that are able to detect low damage levels as well as discrete cracks. Research is also needed to improve the efficiency of the tendons assemblies themselves, including enhancing the properties of the SMP sleeves used in the hybrid tendons

The cradle to gate LCA highlights the dominant contributions of both cement and PET production to the environmental costs, with reference to the global warming and human toxicity environmental categories, which means that kSMP concrete was shown to have a higher environmental impact than conventional concrete. This result was expected, since the assessment did not include the whole life cycle costs. Thus, the benefits associated with reduced maintenance in terms of environmental costs were not reflected in the findings.

## Figures and Tables

**Figure 1 polymers-14-00933-f001:**
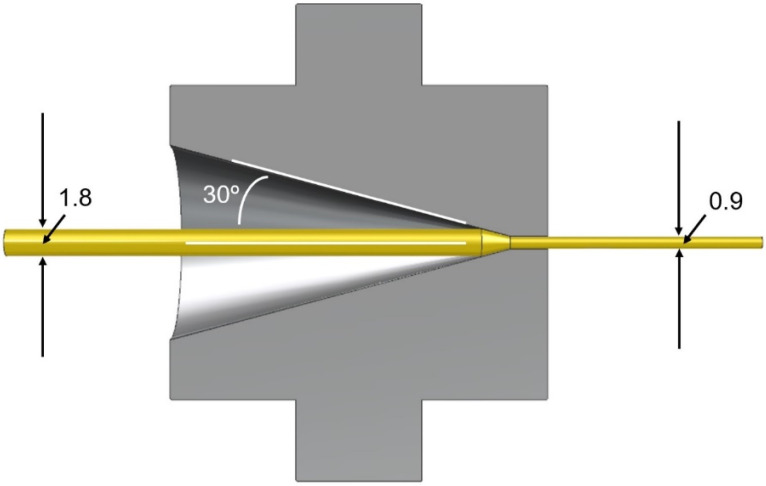
Cross-section of die with oriented fibre emerging on the right. Dimensions in mm.

**Figure 2 polymers-14-00933-f002:**
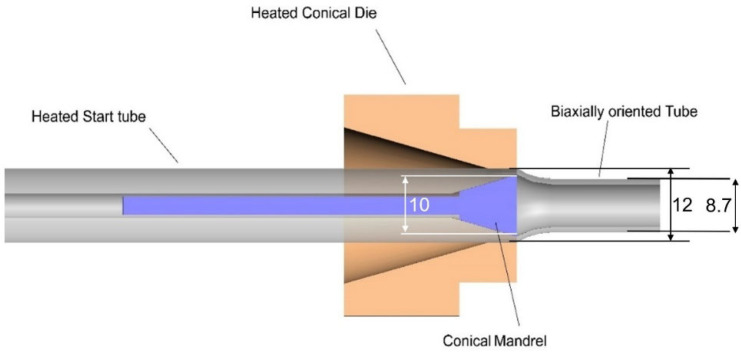
Die arrangement for tube manufacture. Dimensions in mm.

**Figure 3 polymers-14-00933-f003:**
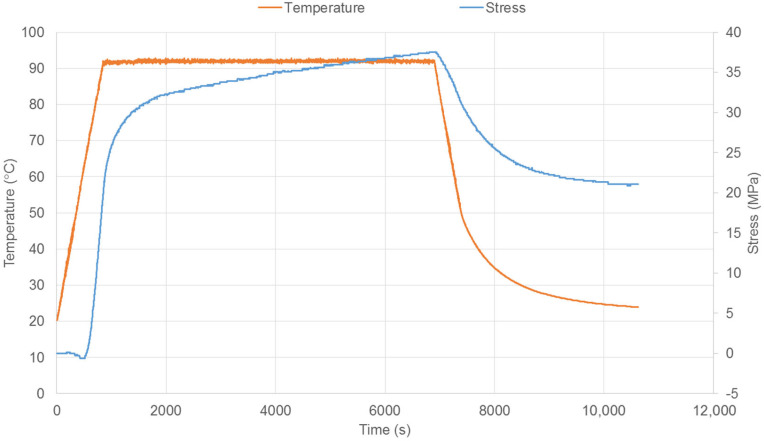
Evolution of shrinkage restraint stress for a single PET fibre.

**Figure 4 polymers-14-00933-f004:**
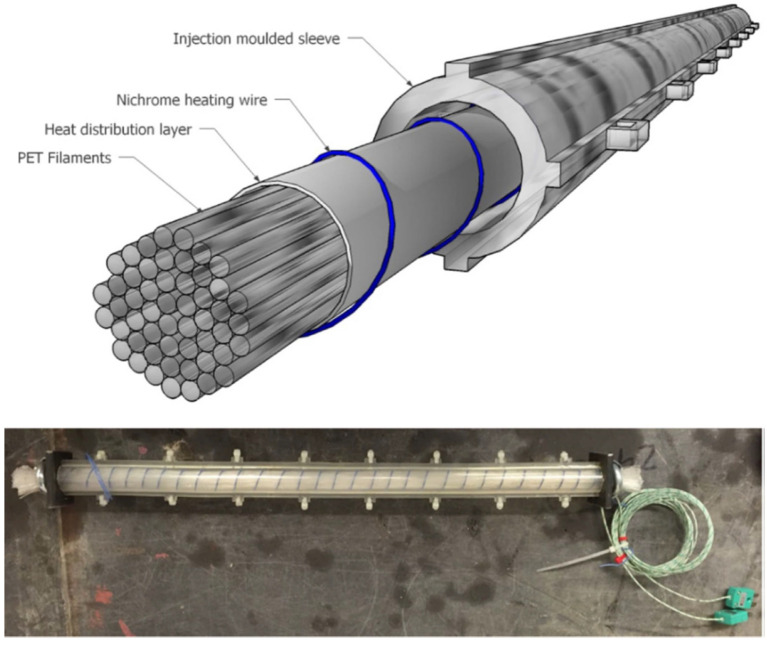
SMP tendon construction, adapted from [42].

**Figure 5 polymers-14-00933-f005:**
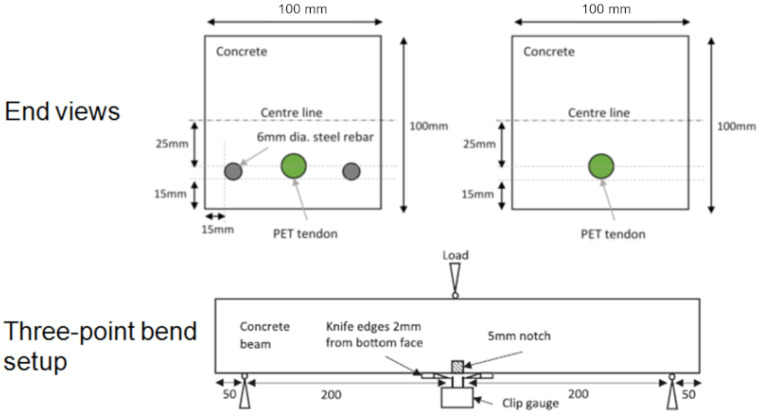
Three-point bend setup and location of tendons, all dimensions in mm, adapted from [42].

**Figure 6 polymers-14-00933-f006:**
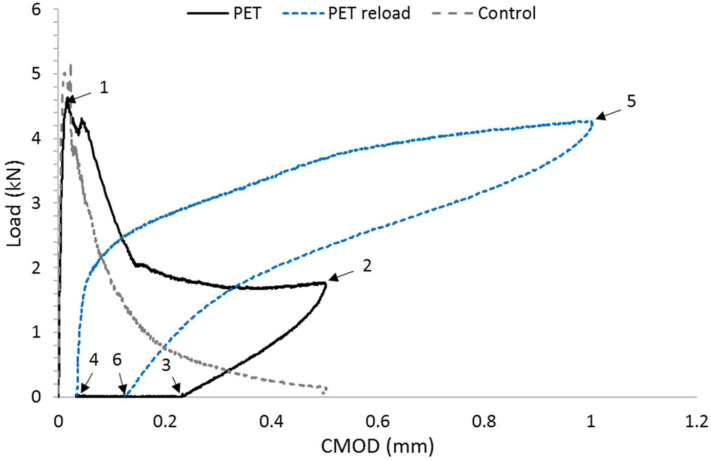
Three-point bend results comparing beams with tendons (PET) and plain-concrete control beams. Initial loading (1–2), unloading (2–3), activation (3–4), reloading (4–5) and unloading (5–6). Adapted from [42].

**Figure 7 polymers-14-00933-f007:**
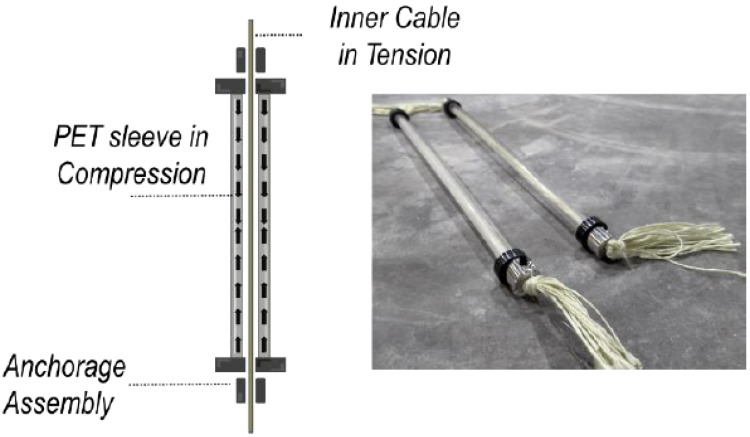
Hybrid device using inner pre-tensioned Kevlar strip (red) and SMP PET outer tube. Adapted from [44].

**Figure 8 polymers-14-00933-f008:**
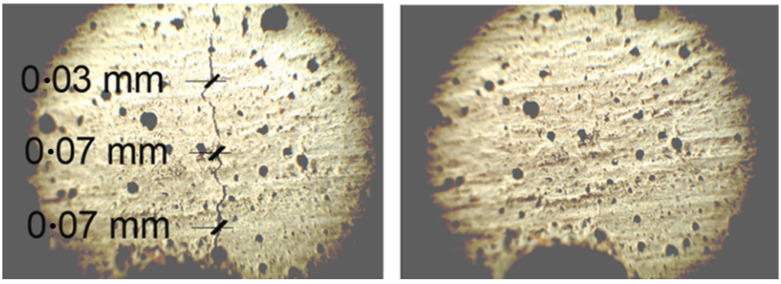
Crack width measurements before and after the hybrid tendons activation. Adapted from [43].

**Figure 9 polymers-14-00933-f009:**
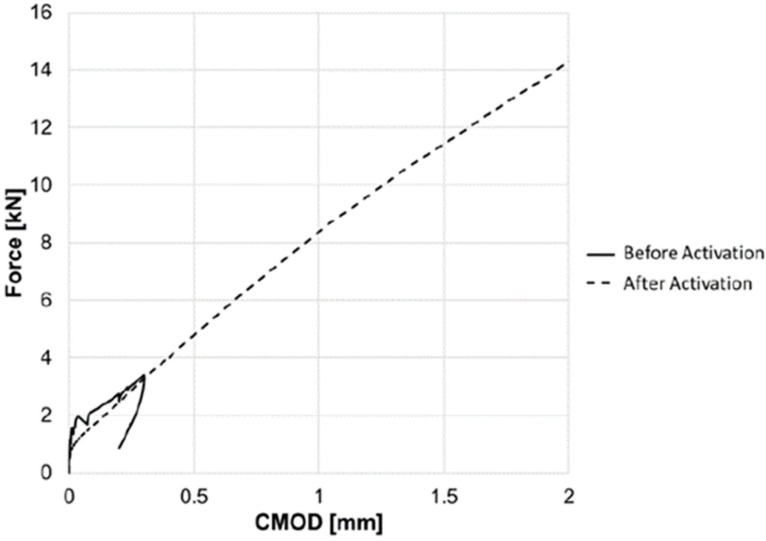
Load vs. CMOD response from flexural beam tests. Adapted from [44].

**Figure 10 polymers-14-00933-f010:**
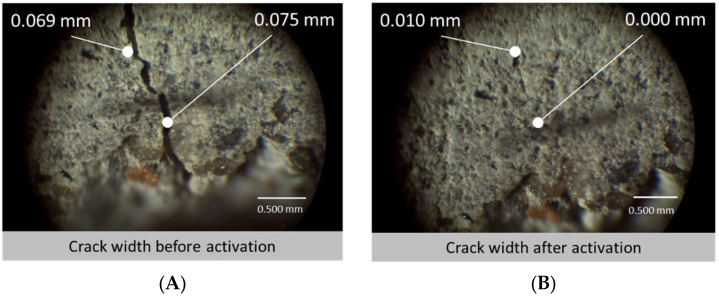
Microscope image of a crack in cementitious sample with knotted shape memory polymers fibres in aligned distribution (AD) before activation (**A**) ad after activation (**B**), adapted from [45].

**Figure 11 polymers-14-00933-f011:**
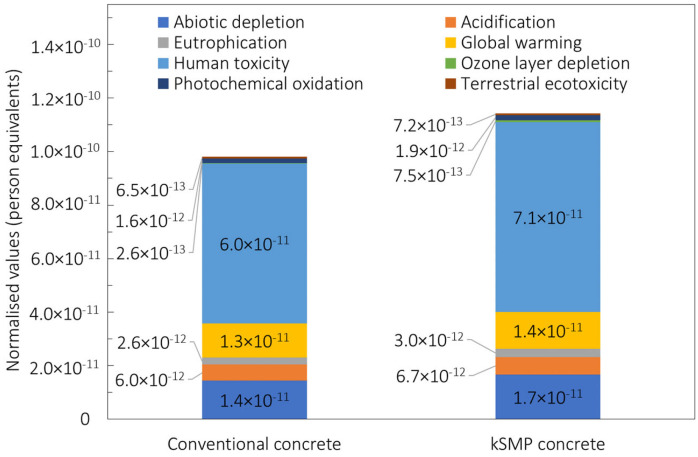
Comparison of LCIA results of the conventional concrete and kSMP concrete (normalization: world, year 2013 CML-2001 person equivalents).

**Figure 12 polymers-14-00933-f012:**
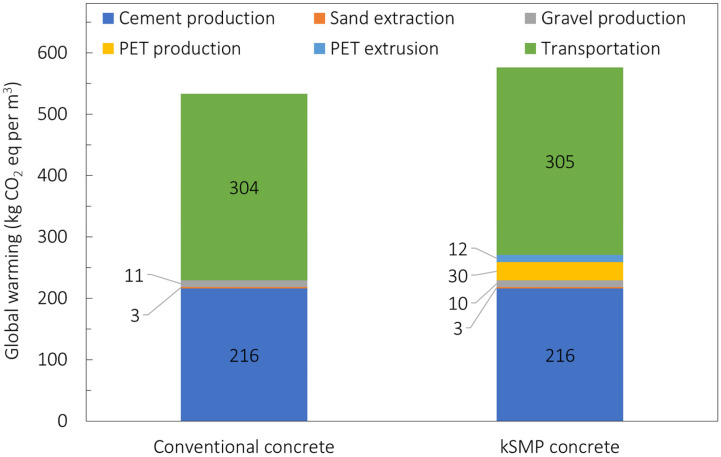
Comparison of normalised values of global warming potential for conventional concrete and kSMP concrete.

**Figure 13 polymers-14-00933-f013:**
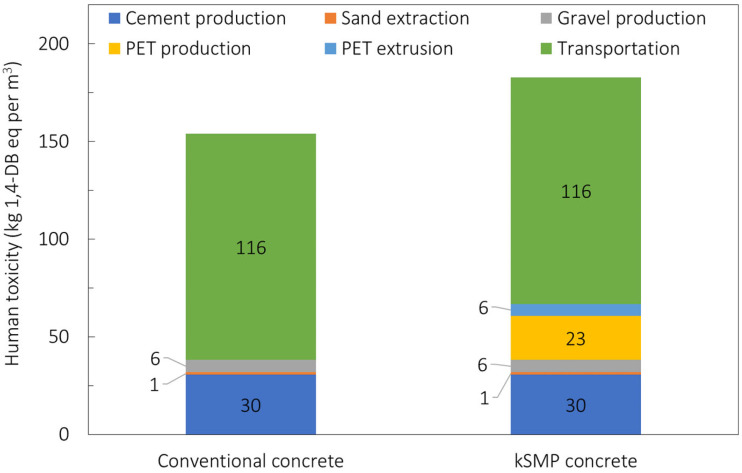
Comparison of normalised values of human toxicity for conventional concrete and kSMP concrete.

**Table 1 polymers-14-00933-t001:** Input to the inventory analysis for the conventional concrete and kSMP concrete.

Direct Burdens, for 1 m^3^ of Concrete	Conventional Concrete	KSMP Concrete
Portland Cement (PC), kg	320	320
Aggregates, kg	1280	1269
Sand, kg	640	640
Water, kg	160	160
K-SMP, kg	0	11
Transport, km	250	250 + 420

**Table 2 polymers-14-00933-t002:** Principal normalized impacts for Conventional concrete and kSMP Concrete.

	Abiotic Depletion	Global Warming	Human Toxicity
Conventional concrete	1.4 × 10^−11^	1.3 × 10^−11^	6.0 × 10^−11^
kSMP concrete	1.7 × 10^−11^	1.4 × 10^−11^	7.1 × 10^−11^

## Data Availability

Information about the data underpinning the results presented here, including how to access them, can be found in the Cardiff University data catalogue at http://doi.org/10.17035/d.2022.0153490818.

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
