# Peer review of "Applications and Life Cycle Assessment of Shape Memory Polyethylene Terephthalate in Concrete for Crack Closure"

_polymers, 2022, doi:10.3390/polym14050933_

Round 1

Reviewer 1 Report

The article is devoted to the study of the properties of shape memory polymer (SMP) products developed for use as concrete strengthening devices. This direction is of scientific and practical interest, however, before this article is accepted for publication, the authors should answer a number of questions that arose when reading this work.

1. In the abstract, the authors should describe in more detail the results of the study, in particular, reflect the numerical values ​​of improved performance in comparison with conventional concrete.
2. The introduction should be significantly expanded, point out the main prospects for the use of polymers for hardening, and also give examples for comparison.
3. What is the reason for such a high temperature range of tests, where it will be used, the authors should explain this experiment.
4. The authors should explain the mechanisms of crack initiation in concrete.
5. In the LCIA results presented, the authors should explain the reason for the increase in toxicity for the concrete type kSMP.

Author Response

Reviewer 1

The article is devoted to the study of the properties of shape memory polymer (SMP) products developed for use as concrete strengthening devices. This direction is of scientific and practical interest, however, before this article is accepted for publication, the authors should answer a number of questions that arose when reading this work.

  • In the abstract, the authors should describe in more detail the results of the study, in particular, reflect the numerical values ​​of improved performance in comparison with conventional concrete.

The abstract has been expanded to cover these points.

  • The introduction should be significantly expanded, point out the main prospects for the use of polymers for hardening, and also give examples for comparison.

The introduction section has been expanded to include the reviewer’s suggestion.

  • What is the reason for such a high temperature range of tests, where it will be used, the authors should explain this experiment.

The origin of the temperature is explained in a new paragraph at the beginning of section 2.2.

  • The authors should explain the mechanisms of crack initiation in concrete.

A paragraph has been added to reflect the reviewer’s suggestion.

  • In the LCIA results presented, the authors should explain the reason for the increase in toxicity for the concrete type kSMP.

The increase in toxicity in the concrete type kSMP is due to the production of the PET as it is reported in the text (379). We have further specified the reason of this increase in the sentence in red“ The dominant role of the cement production is evident from previous figures with the production of the PET which makes the kSMP concrete a solution with higher environmental impacts compared to the conventional one. “This is primarily due to the energy consumption associated with the PET production.”

Reviewer 2 Report

This paper introduce the applications of shape memory PET in structural engineering, the following points can be addressed.

  1. In the introduction part, as the authors stated, the surface and internal cracks are inevitable during the service life of a concrete structure. Therefore, there are many methods can be used to reduce the fibers, including the usage of fibers and expensive agents. I think the authors should enrich the statement of the introduction and tell the potential readers there are also other kinds of methods in addition to the usage of fibers. Also, the existing fiber-related studies should be mentioned. Hence, some related references can improve the introduction part, such as  Comparison of fly ash, PVA fiber, MgO and shrinkage-reducing admixture on the frost resistance of face slab concrete via pore structural and fractal analysis; Influence of MgO on the hydration and shrinkage behavior of low heat Portland cement-based materials via pore structural and fractal analysis.
  2. It is a little confused whether this paper is a review work or an investigation article, since Section 3 is a review work and Section 4 seems like a study. If this is an investigation article, please consider remove the review work (e.g. Section 3) to the introduction part, then list the main results and discussion. Also, in the Conclusion part, it contains many investigation results, such as “it may be concluded that the incorporation of PET-based SMP devices into cementitious structural elements provides an effective means of closing cracks that form after casting and curing”, “…upon thermal activation in both aligned and random distribution, showing crack width reduction up to 77%”. I am really confused whether these conclusions are obtained from the review work or from their own study. So, my simple suggestion is that the authors should exhibit their own investigation work more clearly.
  3. The current title could not cover the main findings of this study, since Section 4 is also an important part.

Author Response

Reviewer 2

This paper introduce the applications of shape memory PET in structural engineering, the following points can be addressed.

  • In the introduction part, as the authors stated, the surface and internal cracks are inevitable during the service life of a concrete structure. Therefore, there are many methods can be used to reduce the fibers, including the usage of fibers and expensive agents. I think the authors should enrich the statement of the introduction and tell the potential readers there are also other kinds of methods in addition to the usage of fibers. Also, the existing fiber-related studies should be mentioned. Hence, some related references can improve the introduction part, such as  Comparison of fly ash, PVA fiber, MgO and shrinkage-reducing admixture on the frost resistance of face slab concrete via pore structural and fractal analysis; Influence of MgO on the hydration and shrinkage behavior of low heat Portland cement-based materials via pore structural and fractal analysis.

A paragraph has been added to reflect the reviewer’s suggestion, with references duly added and discussed.

  • It is a little confused whether this paper is a review work or an investigation article, since Section 3 is a review work and Section 4 seems like a study. If this is an investigation article, please consider remove the review work (e.g. Section 3) to the introduction part, then list the main results and discussion. Also, in the Conclusion part, it contains many investigation results, such as “it may be concluded that the incorporation of PET-based SMP devices into cementitious structural elements provides an effective means of closing cracks that form after casting and curing”, “…upon thermal activation in both aligned and random distribution, showing crack width reduction up to 77%”. I am really confused whether these conclusions are obtained from the review work or from their own study. So, my simple suggestion is that the authors should exhibit their own investigation work more clearly.

All the work reported is that of the current authors, so it is not a review paper. We have included some new words at the end of the introduction and a new subsection that introduces section 3, which clarify the relationship between this paper and previous papers by us in more civil engineering oriented journals where the polymer aspects were less emphasised.

  • The current title could not cover the main findings of this study, since Section 4 is also an important part.

The title has been modified to reflect the reviewers’ suggestion

Reviewer 3 Report

The paper is interesting but still need some major improvements.

  1. Please modify the title and remove the term "Structural Engineering". The use of PET is in concrete, so be specific in title.
  2. Introduction is not systematic. Please make 3-4 systematic paragraphs.
  3. Add paragraph about life cycle assessment to assess the carbon footprint in the introduction.
  4. Add significance of current research in the last paragraph of introduction.
  5. What was the reason for selecting PET fiber. As there are many other fiber's as well. 
  6. Explain the advantage and disadvantages of PET fiber compared to other polymer fibers.
  7. Please consider copyright issue of figures.
  8. Add a discussion before conclusion regarding the practical use/implementation of current study.
  9. What are future suggestion related to current work.
  10. References are too old. Please add latest studies in the introduction.

Author Response

Reviewer 3

The paper is interesting but still need some major improvements.

  • Please modify the title and remove the term "Structural Engineering". The use of PET is in concrete, so be specific in title.

The title has been modified accordingly.

  • Introduction is not systematic. Please make 3-4 systematic paragraphs.

The introduction has been reorganised following reviewer’s suggestion

  • Add paragraph about life cycle assessment to assess the carbon footprint in the introduction.

A paragraph has been added to reflect the reviewer’s suggestion.

  • Add significance of current research in the last paragraph of introduction.

A paragraph has been added to reflect the reviewer’s suggestion.

  • What was the reason for selecting PET fiber. As there are many other fiber's as well. 

The considerations governing the choice of polymer are given in a new paragraph at the beginning of section 2.

  • Explain the advantage and disadvantages of PET fiber compared to other polymer fibers.

We have addressed this in the new paragraph at the start of section 2.2.

  • Please consider copyright issue of figures.

The figures reported are adapted from previous work published by the authors and reference is given.

  • Add a discussion before conclusion regarding the practical use/implementation of current study.

A paragraph has been added to reflect the reviewer’s suggestion.

  • What are future suggestion related to current work.

A paragraph has been added to reflect the reviewer’s suggestion.

  • References are too old. Please add latest studies in the introduction.

The papers on the invention and development of die drawing are “old” since they reflect the time at which the events happened. Two of them have been replaced by descriptions of more recent developments.

Round 2

Reviewer 1 Report

The authors answered all questions according to the comments of the reviewer. The article may be accepted for publication.

Reviewer 2 Report

The authors did a good revision work, I suggest acceptance of this paper.

A small place should be revised during the proofreading stage that, the "LCA" in the title should be given its full name.

Reviewer 3 Report

Accept in current form.